# STRAINS: A big data method for classifying cellular response to stimuli at the tissue scale

**Jingyang Zheng**[1]*, **Thomas Wyse Jackson**[1], **Lisa A. Fortier**[2], **Lawrence J. Bonassar**[3,4], **Michelle L. Delco**[2], **Itai Cohen**[1]

**1** Department of Physics, Cornell University, Ithaca, NY, United States of America, **2** College of Veterinary Medicine, Cornell University, Ithaca, NY, United States of America, **3** Meinig School of Biomedical Engineering, Cornell University, Ithaca, NY, United States of America, **4** Sibley School of Mechanical and Aerospace Engineering, Cornell University, Ithaca, NY, United States of America

* jz848@cornell.edu

## Abstract

Cellular response to stimulation governs tissue scale processes ranging from growth and development to maintaining tissue health and initiating disease. To determine how cells coordinate their response to such stimuli, it is necessary to simultaneously track and measure the spatiotemporal distribution of their behaviors throughout the tissue. Here, we report on a novel SpatioTemporal Response Analysis *IN Situ* (STRAINS) tool that uses fluorescent micrographs, cell tracking, and machine learning to measure such behavioral distributions. STRAINS is broadly applicable to any tissue where fluorescence can be used to indicate changes in cell behavior. For illustration, we use STRAINS to simultaneously analyze the mechanotransduction response of 5000 chondrocytes—over 20 million data points—in cartilage during the 50 ms to 4 hours after the tissue was subjected to local mechanical injury, known to initiate osteoarthritis. We find that chondrocytes exhibit a range of mechanobiological responses indicating activation of distinct biochemical pathways with clear spatial patterns related to the induced local strains during impact. These results illustrate the power of this approach.

## Introduction

To sustain tissue function, cells must coordinate their response to external stimuli. In mechanically sensitive tissues, for example, cells sense their environment using cytoskeletal elements, ion channels, and other mechanisms to initiate such coordinated behaviors [1, 2]. Changes in fluid pressure in vascular systems affect mechanosensitive ion channels, driving cell migration and muscle development [3]. Altered mechanosensing in cancer cells makes them unable to sense stiffness, potentially playing a role in metastasis, migration, and disease progression [4]. And, in cartilage, tendon, and bone, cell mechanosensing pathways regulate growth and development during normal function or promote disease during aberrant loading [5–7]. Pioneering studies furthered our understanding of which mechanotransduction pathways are activated in *single* cells in response to various perturbations [8]. Studies using pillar arrays [9–11], traction force microscopy [12], magnetic tweezers [13], or optical traps [14] for example, have

**Data Availability Statement:** The data can be found at the Cornell eCommons repository at: https://doi.org/10.7298/3kwt-pm43 The code can

be found on Github at: https://github.com/jingyangzheng/STRAINS.

**Funding:** Funding: The work was supported by the NIH National Institute of Arthritis and Musculoskeletal and Skin Diseases (niams.nih.gov), Contract: 5R01AR071394 to LAF, K08AR068470 to MLD, R03AR075929 to MLD, and The Harry M. Zweig Fund for Equine Research to LAF and MLD. Additionally, this work was supported by the National Science Foundation (nsf.gov) grants DMR-1807602 to IC and LJB, CMMI-1927197 to LJB and IC, and BMMB-1536463 IC and LJB. Lastly, this work made use of the Cornell Center for Materials Research Shared Facilities which are supported through the NSF MRSEC program (DMR-1719875). The funders had no role in study design, data collection and analysis, decision to publish, or preparation of the manuscript.

**Competing interests:** The authors have declared that no competing interests exist.

demonstrated the role of substrate rigidity in stem cell differentiation [15], the alignment of cellular microfilaments in the direction of force [16], and the highly varying force profiles of migrating cells [8, 17]. Such cellular responses must be coordinated at the *tissue scale* to sustain mechanical function, direct resources to regions in need of repair, or initiate healing [18–20]. This coordination, however, remains poorly understood because few techniques are available for imaging, analyzing, and sorting the *in situ* collective response of thousands of cells over thousands of time points throughout the tissue.

Here, we introduce a SpatioTemporal Response Analysis IN Situ (STRAINS) tool that uses new experimental methods and a big data analysis technique to investigate tissue scale coordination of the cellular responses. STRAINS tracks thousands of cells within tissue during and after an applied stimulation, extracts their individual fluorescence traces, and analyzes their spatiotemporal behavior patterns. This technique makes use of newly developed protocols to stain and image processes such as $Ca^{2+}$ signaling, mitochondrial depolarization, and nuclear membrane permeability *in situ* over sub-second to hour time scales. The advances we report here entail tracking responses in thousands of cells that are moving, visualizing millions of data points with an intuitive graphical user interface (GUI), and using new custom sorting and machine learning algorithms to classify and map a wide range of cellular behaviors throughout the tissue.

We demonstrate the utility of this approach by using STRAINS to investigate the complex relationships between mechanical strain and chondrocyte responses in articular cartilage, identifying distinct patterns of cell behaviors and mapping their spatiotemporal distribution. Macroscale joint injury, specifically rapid cartilage overloading, is known to precipitate osteoarthritis.

For example, previous work has demonstrated that articular impact injury triggers tissue scale catabolic responses *in situ* and *in vivo* [21–25]. During impact, chondrocytes use mechanosensors like integrins [26, 27], the primary cilium [28–31], and various mechanosensitive ion channels [9, 32–35] to convert mechanical signals into biochemical responses ranging from the synthesis of extracellular matrix proteins for maintaining tissue integrity to apoptosis and matrix degradation.

Signaling within and between cells in cartilage post impact occurs on multiple timescales. Within seconds, activation of mechanosensors on the cellular membrane enables calcium and other force-sensitive signaling [32, 36]. In the ensuing hours, the initial cellular response affects mitochondrial polarization [23], cell viability [22], and subsequent signaling cascade, leading to distinct outcomes based on the initial local strain experienced by the cell. By measuring both short (sub-second to second) and long (minutes to hours) term signaling, STRAINS enabled us to make connections between signaling events and paint a fuller picture of the signaling landscape after injury.

Additionally, the complex structure of articular cartilage matrix causes local strain within the tissue to vary with location and depth, which in turn can lead to different cellular behaviors. For example, cells directly below the impact site primarily experience compression. In contrast, cells to the sides of the impact site experience greater shear stresses. In single cells, these distinct mechanical deformation modes are known to trigger different responses in the chondrocytes [37]. Whether cells maintain these individual behaviors based solely on the local deformations they experience or coordinate their response more globally is poorly understood.

Collectively, cartilage's depth-dependent spatial heterogeneity, the complex load distribution within the tissue during impact, and the broad range of timescales for chondrocyte responses makes it an ideal tissue for showcasing the power of this method. Importantly,

however, STRAINS can be applied to any system where it is relevant to study the collective spatiotemporal response of large numbers of cells to external stimuli.

## Results

### Experimental system and *in situ* imaging procedure

We have developed a microscale impacting system to assess the real-time multichannel cellular response to mechanical stimulus. A custom-built confocal-mounted impactor was used to injure and image fluorescently stained bovine cartilage samples [21]. Specifically, 6 mm plugs were sterilely extracted from the condyles of neonatal bovids (Fig 1a). Each plug was bisected into two hemicylinders, cultured for stabilization [24] and stained for simultaneous measurement of $Ca^{2+}$ concentration (Calbryte 520 AM), mitochondrial depolarization (tetramethylrhodamine, TMRM), and nuclear membrane permeability (Sytox Blue) (Fig 1b, and Methods). To image the tissue response, two hemicylinders from the same plug were glued onto the fixed backplate of the confocal-mounted impactor (Fig 1c). One sample was used as a control while the second sample was impacted to induce injury. The impactor was calibrated to deliver to the articular surface a 5–10 ms impact with a peak stress of ∼1 MPa, which produced superphysiologic strains and strain rates encompassing the wide range of strains observed in joints with traumatic injury within a small field of view [22, 23]. Strain calibration was conducted with a high speed camera (v7.1, Vision Research), as described in Bartell et. al., 2015 [21] and Henak et. al., 2017 [38]. To create visual texture for measuring strain fields, the cut surface of the impacted sample was coated with fluorescent polystyrene microspheres (2 μm carboxylate particles). During impact, cartilage deformation was recorded using the high speed camera at

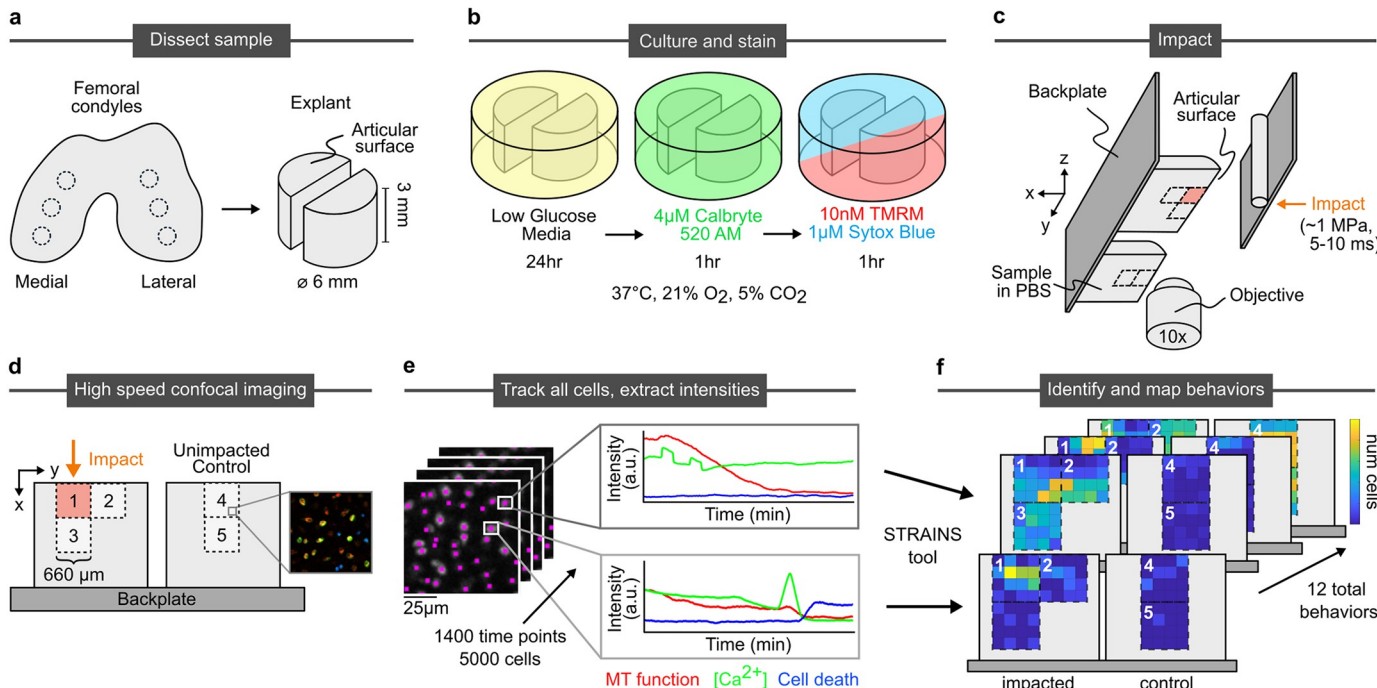

**Fig 1. Sample processing, imaging, tracking, and intensity analysis procedure.** a) Biopsy punches of condylar articular cartilage are collected and bisected. b) Hemicylinders are cultured and stained for calcium concentration, mitochondrial polarization, and nuclear membrane permeability (stand in for cell death). c) Samples are mounted side-by-side to the back plate of impactor. One half of the sample impacted once, with the other serving as an unimpacted control. d) Imaging occurs during impact at site 1 and after impact at sites 1–5. e) Individual cells are tracked through all time points and their stain intensities extracted. f) Temporal patterns of strain intensities are classified by STRAINS into distinct categories of cell behaviors, which are identified and mapped onto tissue location.

1000 frames per second. Digital image correlation was used to track the deformation and calculate strain fields.

This entire apparatus was loaded onto a fast confocal microscope that enabled imaging of the cells throughout the impact and relaxation process. Each region corresponding to the field of view for our 10X objective was 660 μm x 660 μm in size. We assessed multiple regions to understand the influence of a wide range of local tissue strains on the behaviors of cells (Fig 1d). On the impacted sample, we imaged the entirety of the impact site (region 1) at the surface of the tissue from 60 μm above the articular surface to 600 μm below the surface (superficial zone of articular cartilage), lateral to and at the same depth as the impact site (region 2), also at the surface of the tissue, and directly below the impact site (region 3), in the region extending 600 μm to 1260 μm below the surface (middle zone of articular cartilage). On the unimpacted control, we imaged two sites (regions 4 and 5) at the same tissue depths (superficial and middle zones) as the impacted sample in order to compare cells of similar phenotype. From this imaging process, we obtained the fluorescent intensities of each cell (expanded image in Fig 1d). For region 1 we imaged the $Ca^{2+}$ response at 40 frames per second over 1.5 min during and immediately after the impact. Subsequently, we imaged all 5 regions and all three color channels every 10 seconds over a 4 hour period. Collectively, we obtained the time-dependent fluorescent response of each channel for ~5000 cells corresponding to ~20 million data points (Fig 1e). Each cell exhibited a pattern of intensity responses with time for the three fluorescent channels. Once classified, the location and frequency of these distinct temporal response patterns within the tissue were mapped (Fig 1f).

## Strain-dependent cellular response

The strain field resulting from impact and the associated cell response had complex behaviors that varied spatiotemporally. Specifically, we observed complex patterns in the immediate post-impact $Ca^{2+}$ response and hours-long cellular responses for all three measured signals. In the milliseconds after impact trauma, increased calcium concentration can be observed in cells proximal to the impact site, in the superficial region. However, on the seconds timescale, we observed differences in total intensity between chondrocytes experiencing shear and compression within this region (Green in Fig 2a). On longer time scales, we found that mitochondrial polarity rapidly diminished at the impact site in the minutes after injury, with calcium concentration following the same pattern but with some cells exhibiting transients on the scale of minutes (S1 Video). Conversely, nuclear membrane permeability initially showed a very low intensity throughout the region and reached higher intensities in a fraction of the cells in regions extending up to 400 μm below the impact site on a time scale of hours. Consistent with the short time calcium response, this pattern of cell death did not extend to areas of the tissue which experienced primarily shear strains.

Collectively, these distinct spatiotemporal patterns of cell response indicated that multiple mechanobiological pathways may have been activated in response to local strain. Developing an understanding of how such processes are related requires identifying distinct cellular signatures and mapping out where in the tissue they are localized. To obtain these maps, however, we must first identify each cell, track its movement and multi-channel fluorescence response over time (Fig 1e), and classify its cellular signature (Fig 1f).

## Enhanced particle tracking captures behaviors of moving cells

In order to measure tissue-level cellular behaviors, we must first track each cell individually, over time and through movement, to obtain the fluorescence intensities of each channel. To track the cells, we summed the intensities from all three fluorescent channels at each time step

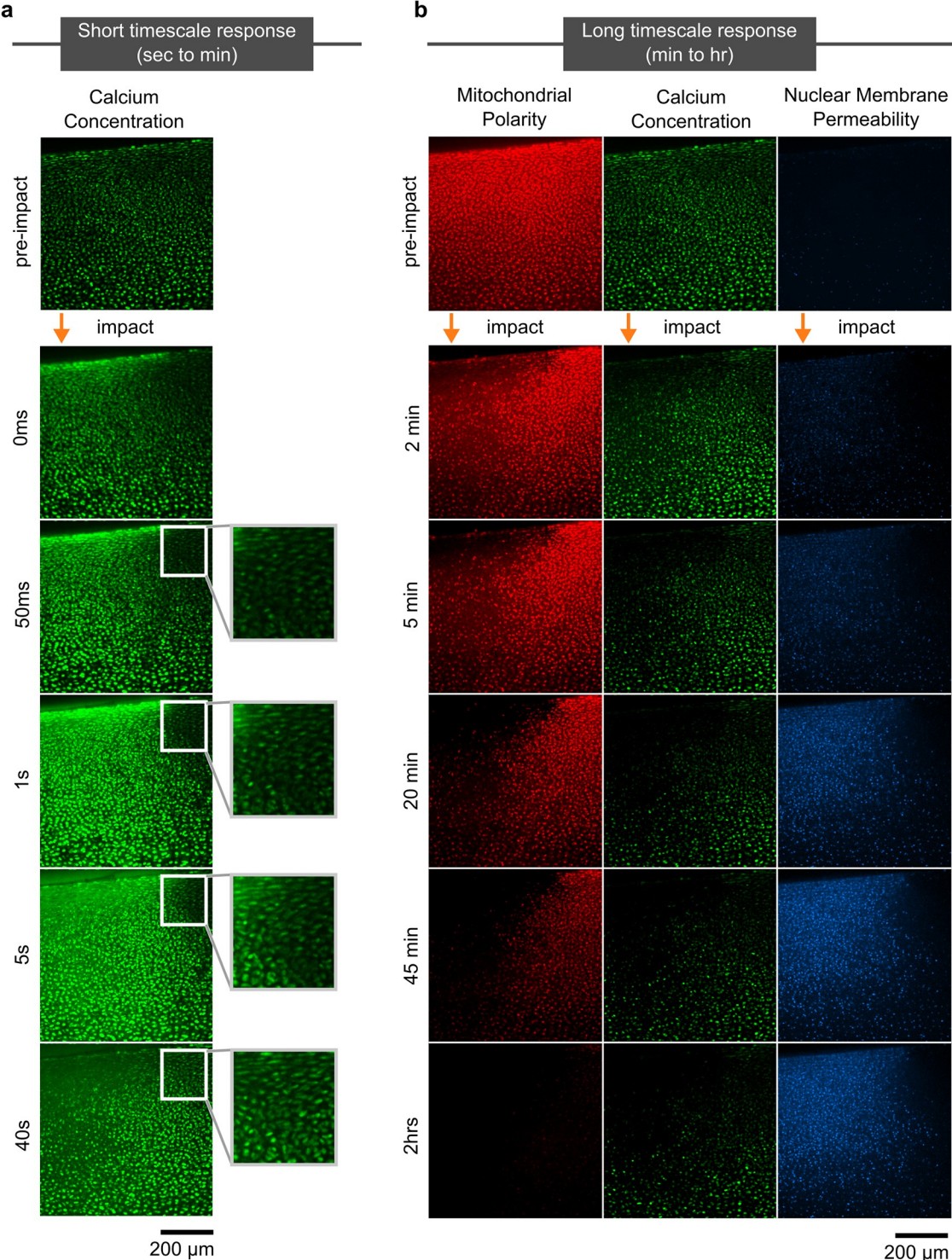

**Fig 2. Timelapse of cartilage response.** Images are shown at the impact site (region 1), with the impact location labeled with an orange arrow. a) High speed imaging of calcium concentration in the minute after impact showed a wave of calcium uptake in cells moving outward from the immediate impact site. Inset shows the area experiencing mostly shear strain, with a more muted and delayed response in comparison with areas experiencing compressive strain. b) Long term imaging of all three stains (mitochondrial polarization, calcium concentration, nuclear membrane permeability). As time progressed, mitochondrial polarization and calcium concentration slowly decreased while nuclear membrane permeability progressively increased with greatest intensity within 400 μm of the impact site.

such that the composite image showed bright isolated regions corresponding to the cells. We then applied a modified version of Crocker and Grier's particle tracking algorithm to filter the image and obtain trajectories for each cell centroid (Fig 1e) [39]. Using this method, we obtained the fluorescent time traces of over 5000 cells for the five imaged regions yielding over 20 million measurements.

Importantly, the sheer scale of data made it prohibitively difficult to use typical statistical analyses or cell tracking tools, which are limited in tracking capabilities and unable to pick out specific time series characteristics and interactions between multiple different fluorescent channels tracking different components of cellular function. In order to interpret our data, we needed to categorize cellular signatures, relate each cells' response to its location within the tissue, and determine whether the cellular signals from multiple cells were spatiotemporally clustered.

## A MATLAB graphical user interface enables identification of cell behaviors

To address this challenge, we built a graphical user interface (GUI) in MATLAB to allow researchers to directly make comparisons between images and time series signatures (Fig 3). A built-in video player allows the user to scan through and select image frames for analysis. Individual cells within any frame can be selected by clicking on the image, entering the cell ID (the number assigned by the particle tracking algorithm), or providing its x-y pixel coordinate (the

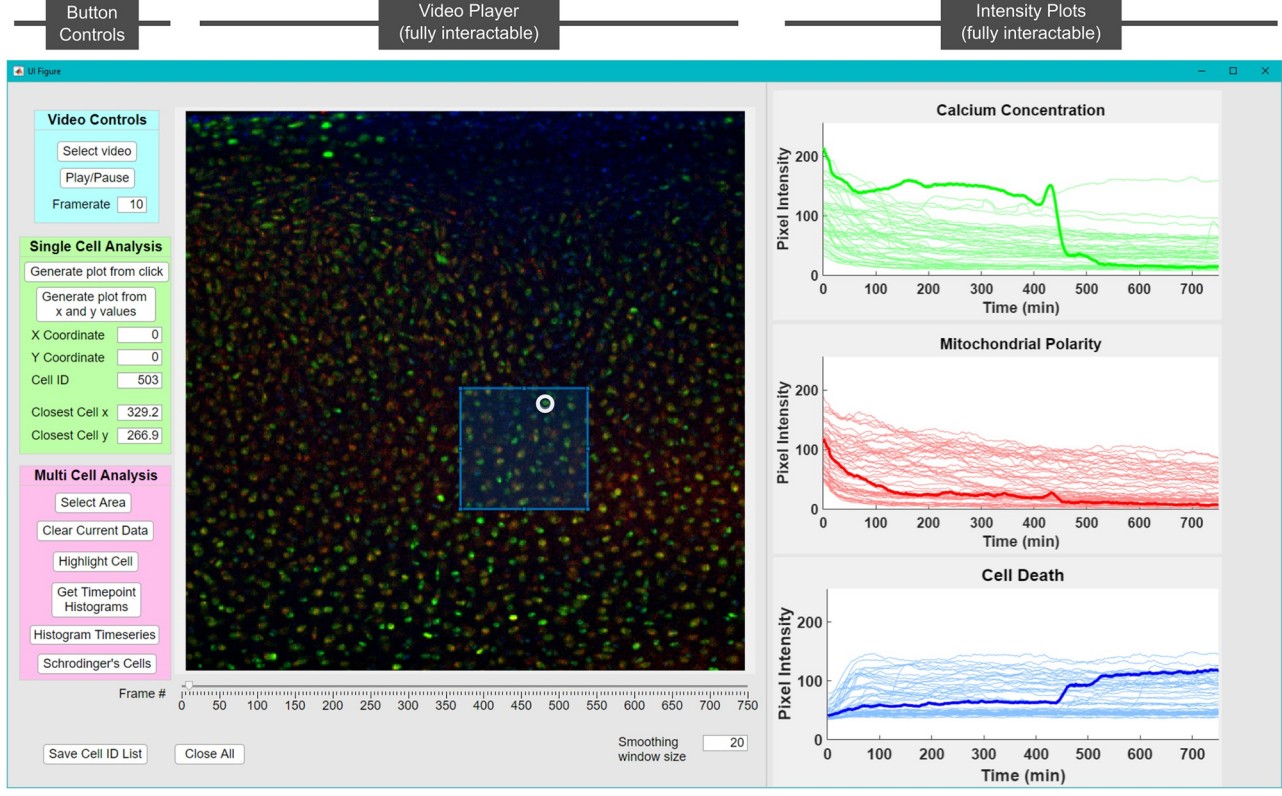

**Fig 3. A MATLAB GUI for image processing.** Shown on the left is a panel of controls for video selection and scanning, single cell analysis, and multi cell analysis. Shown in the center is the video player which allows for choosing specific frames or time points using a slider. The GUI allows selecting and displaying data for either individual cells or regions within the frame. On the right are the resulting plots showing channel intensities. The plots can also be clicked on to highlight cells of interest within a group (which will draw a circle around the cell in the image). In addition, time points can be selected to generate cross-sectional histograms for each stain. A full video detailing GUI functionality can be found in the S1 Video.

program will find the nearest cell). The program then plots the three color intensity versus time curves for that cell on the right side of the GUI. To enable the analysis of multiple cells, our GUI allows a user to select a rectangular region within any frame, and the program will plot the fluorescence curves for all the cells within the region. Directly clicking on a time series in one of the plots will bold the selected line in each color and circle the selected cell within the image (Fig 3). Finally, the cell ID of all observed cells can be saved to a text file before exiting the program for record keeping. This GUI allowed for targeted investigations of cellular behaviors in different regions of the tissue.

Using this GUI, we found specific repeating patterns in the intensity curves related to observed peaks (transients), intensity jumps, decay time scales, plateaus, and the temporal locations of such features. For example, cell death was identified as a sudden increase in nuclear membrane permeability (see for example bold blue channel in Fig 3). This behavior often followed a peak in the calcium concentration (green channel in Fig 3). We also used the GUI to distinguish between seemingly similar curves. While numerous cells showed a rapid increase in nuclear membrane permeability within half an hour of impact, in some cells this signal plateaued and remained high, while in others it slowly decayed to a lower plateau. These differences in time series shape are subtle but distinct. Collectively, this GUI and the analysis features it enabled provided a pathway for sorting the millions of data points in an intuitive fashion, enabling the user to quickly identify categories of cell behaviors and develop an intuition for where each behavior tends to localize.

Making use of the GUI analysis features, we identified twelve distinct behaviors across all cells within an impacted tissue and mapped their location relative to the impact site (Fig 4). We observed eight different behaviors where chondrocytes showed a high level or a rapid increase in their nuclear membrane permeability, likely related to cell death (Fig 4a–4g and 4l). Additionally, two behaviors were related to distinct calcium transients (Fig 4h and 4i). Finally, we identified two behaviors where cells maintained low nuclear membrane permeability throughout the experiment (Fig 4j and 4k).

Importantly, the precision with which we measure the cell intensities and correlations in cell signatures allowed for distinguishing between these behaviors, even when the accompanying curves were quite similar. For example, in some cells the calcium transients were not associated with an increase in nuclear membrane permeability (Fig 4i) while in others, the calcium transient occurred immediately before a rise in nuclear membrane permeability (Fig 4h) or at some time after the nuclear membrane permeability rose (Fig 4g). The differences in how these two channels interacted, along with the distinct spatial distributions associated with each behavior, indicated that separate biochemical processes associated with activation of calcium channels may have taken place, illustrating the power of our approach for generating a comprehensive map of the tissue scale multi-channel cellular response to an applied perturbation. By quickly identifying and spatiotemporally mapping behaviors ranging from normal to abnormal, our approach allows us to focus on interesting and puzzling behaviors for future experimentation.

## Implementing automated sorting algorithms to identify cell behavior and category

While this analysis framework is clearly very useful for identifying distinct categories of behaviors, it required extensive manual sorting, which is cumbersome for the scale of data acquired from our technique. This burden, however, was lightened by implementing an augmented strategy which combined the manually sorted categories with custom feature extraction algorithms and supervised machine learning to quickly classify thousands of cellular responses in

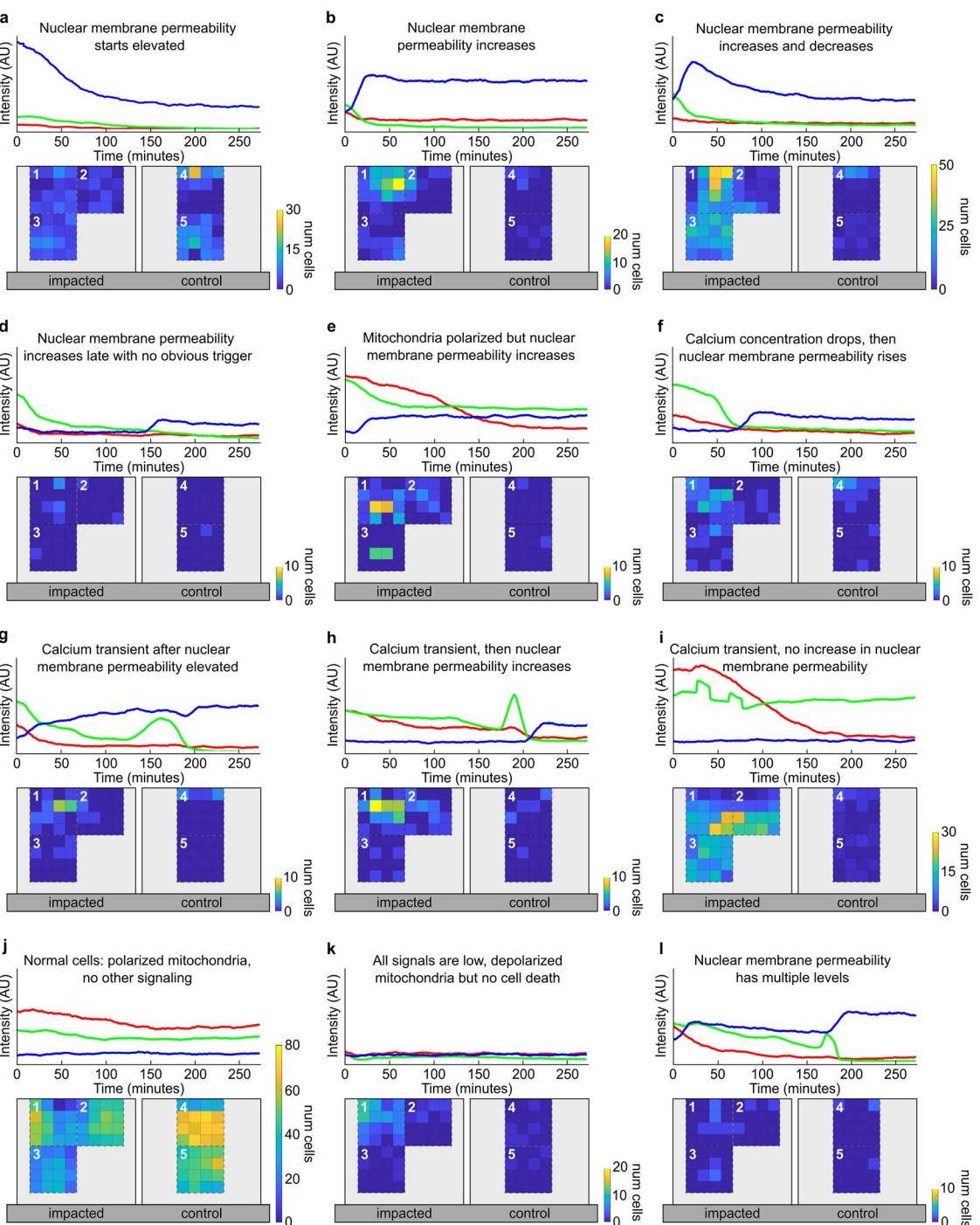

**Fig 4. Cell behavior categories and their distributions.** For each category, a representative time series for a single cell (top, with blue representing nuclear membrane permeability, green representing calcium concentration, and red representing mitochondrial polarity) along with a heat map of cell counts indicating relative frequency at different locations are shown. Colorbar maxima vary between categories to resolve spatial distributions for categories with fewer cells. Regions 1–3 are on the impacted sample, with 1 as the site of impact. Regions 4–5 are on the control sample. a) The nuclear membrane permeability starts out elevated and decays. b) The nuclear membrane permeability increases within 30 min of impact and plateaus or continues increasing. c) The nuclear membrane permeability increases within 30 min of impact and then decays. d) The nuclear membrane permeability increases after 30 min after impact but no prior signaling event is observed. e) The nuclear membrane permeability increases despite the mitochondria being polarized. f) The nuclear membrane permeability increases after the calcium concentration drops after a period of sustained elevation. g) Transient calcium signaling is observed after the nuclear membrane permeability has already increased. h) The nuclear membrane permeability increases after calcium transient(s). i) Calcium transient(s) are observed with no increase in

nuclear membrane permeability. j) No calcium transients or increases in nuclear membrane permeability. k) All three signals are very low. l) The nuclear membrane permeability increases multiple times.

subsequent experiments. While such algorithms are invariably somewhat system specific, it is nevertheless instructive to illustrate their implementation in our system.

**Programmed feature extraction of time series characteristics.** Our first approach to implement cellular behavior classification employed a custom feature extraction and decision tree algorithm. In our system, sudden signaling events like calcium transients (peaks) or sharp changes to cell nuclear membrane permeability (changepoints) played an important role in dictating which category a cell belongs to. These events, however, can occur at random times during the imaging process, making it difficult to search for specific features using automated machine learning methods like clustering or classification. Here, we made use of peak and changepoint detection algorithms in MATLAB alongside extraction of basic time series statistics (minima, maxima, mean, variance, range, etc.) to find the identifying features of each behavior category. The relationship between these specific features and other time series characteristics could then be used to create a "fingerprint" for each behavior, which could be searched for en masse.

Once these features were identified, the fluorescence response for each cell was categorized using a decision tree. The tree started by establishing if a cell has a certain feature, such as a step in the nuclear membrane permeability channel or a calcium transient and then branches to more specific criteria based on the relationships between them. For example, if a cell had a changepoint in nuclear membrane permeability, then the tree moved to more specific criteria such as whether the fluorescence in this channel rose and fell or remained high.

In order to distinguish between these possibilities, we searched for peaks in the blue nuclear membrane permeability signal. If the nuclear membrane permeability showed a peak, then the cell was classified as shown in Fig 4c. A similar process was applied to classify all twelve manually identified behaviors (See Methods).

Using this decision tree, we found that the best classified categories reached ~85% accuracy, defined as the true positives divided by the total number of cells manually identified for that category (Table 1). Using this decision tree were were able to correctly classify the vast majority of cells (4331 out of 5347) with a total accuracy of 81%. Most importantly, while such

**Table 1. Category based accuracy of decision tree classification.**

| Category | Accuracy | Total Cells |
|---|---|---|
| Nuclear membrane permeability starts elevated (Fig 4a) | 0.85 | 313 |
| Nuclear membrane permeability increases (Fig 4b) | 0.56 | 151 |
| Nuclear membrane permeability increases and decreases (Fig 4c) | 0.84 | 748 |
| Nuclear membrane permeability increases late (Fig 4d) | 0.18 | 11 |
| Mitochondria polarized but nuclear membrane permeability increases (Fig 4e) | 0.06 | 63 |
| Calcium concentration drops, then nuclear membrane permeability increases (Fig 4f) | 0.62 | 60 |
| Nuclear membrane permeability elevated, but calcium transient occurs afterwards (Fig 4g) | 0.44 | 41 |
| Calcium transient, then nuclear membrane permeability increases (Fig 4h) | 0.73 | 73 |
| Calcium transient, then no changes to nuclear membrane permeability (Fig 4i) | 0.85 | 524 |
| Normal cells, polarized mitochondria (Fig 4j) | 0.84 | 3244 |
| All signals low, depolarized mitochondria (Fig 4k) | 0.34 | 101 |
| Nuclear membrane permeability has multiple levels (Fig 4l) | 0 | 18 |

**Table 2. Time series classification methods with details on classifier types and their corresponding scores.**

| Classifier | Accuracy |
|---|---|
| Canonical Interval Forest (CIF) [42] | 0.889 |
| Diverse Representation Canonical Interval Forest (DrCIF) [42] | 0.887 |
| RandOm Convolutional KErnel Transform (ROCKET) [43] | 0.852 |
| Arsenal Ensemble [42] | 0.854 |

decision trees must be determined individually for each system, once established, they can be used repeatedly and with high fidelity.

**Time series classification of chondrocyte signaling using sktime library.** A second strategy we used to identify cell signatures was time series classification, a type of supervised machine learning where a model system learns to assign labels to objects based on training examples. In our system, relevant information for our data was embedded as a multi-channel time series, so we made use of multivariate time series classifiers from the Python sktime library [40, 41]. Here, we randomly split the data into training and testing sets and optimized the classifiers over a range of parameters. For example, for the Canonical Interval Forest (CIF) classifier [42], we found that the accuracy plateaued once we used more than 1000 estimators. Similarly, for the RandOm Convolutional KErnel Transform (ROCKET) classifier we found that the accuracy plateaued when we used more than 100000 kernels [43]. For detailed list of parameters, see Methods. Finally, we determined what percentage of cells were labeled correctly by the classifier (Table 2).

We found that classifiers worked better at detecting certain categories. Our system was an imbalanced multi-label classification problem where there were multiple categories of behaviors with unevenly distributed numbers of objects between categories. Since classifiers are much better trained on categories with more cells, the heavily populated behavior categories were identified with much higher accuracy. Also, when cells had specific features that were not temporally consistent, (e.g. calcium transients or changes in nuclear membrane permeability that occurred in varying numbers, with different amplitudes, and at varying time points) classifiers struggled because no two signals were exactly the same. Consequently, we found that higher scores could be achieved when we split the cells with calcium transients from the data set and classified them separately. When combined with the decision tree algorithm described above, these supervised time series classifiers achieved accuracy values of up to 89%.

These results speak to the vast potential for automated sorting in future studies [44]. In particular, since the algorithms used here were only recently developed, it is likely that as new more powerful classifiers become available such supervised machine learning approaches will produce greater sorting accuracy. One could even imagine future implementations where unsupervised machine learning is used to extract the most impactful features of the data and cluster cell behaviors with minimal human effort. More broadly, these results demonstrated that a strategy combining automated feature extraction with various machine learning techniques could effectively sort complex cellular data in a streamlined and automated process.

## Discussion

Using STRAINS to monitor cellular behaviors *in situ*, categorize them, and determine where in the tissue they occur enabled novel observations about mechanotransduction in articular cartilage that could not have been obtained from single cell experiments. For example, high

levels of compressive strain and transiently high hydrostatic pressure are known to dominate close to the impact site [45] and previous work in our group has demonstrated that microscale local strain exceeding 8% causes cell death [22]. However, by using STRAINS to continuously collect cell behavior data after impact, we were able to observe nine categories of behavior associated with increased nuclear membrane permeability, which suggested different pathways to cell death (Fig 4a–4h and 4l). Here, the nuclear membrane permeability increased and decreased (Fig 4c), or increased and plateaued (Fig 4c), indicative of cell death due to super-physiologic strain. Further from the impact site we observed multiple behaviors related to elevated nuclear membrane permeability, but with additional signals suggesting other biological mechanisms at work. For example, a subset of cells displayed multiple levels of nuclear membrane permeability (Fig 4l), which may reflect multiple inputs or different stages of cell death processes. Collectively, such results open the door to analyzing how cellular responses are coordinated at the tissue scale. These empirical observations do not drive a concrete understanding without further experimentation, but provide the foundation for establishing how cellular behaviors change in future experiments aimed at probing specific ion channels, mitochondrial function, senescence, and other processes.

More broadly, STRAINS is customizable for analyzing many other tissue systems, and scales well for large numbers of cells. In particular, while the data shown here relate to investigating mechanotransduction in articular cartilage, the described techniques can be applied to any tissue scale system where cell response can be quantified using fluorescence. For example, traumatic brain injury is caused by large mechanical forces on brain tissue [46], with cellular mechanotransduction playing an important role in pathology [47]. Similar staining protocols could enable application of STRAINS to this system. In cardiac tissue, the role of mechanotransduction in determining cardiac myocyte behavior has been studied in single cells, but STRAINS can be applied to address these questions on a tissue scale [48]. In tumors, drug diffusion is hampered by various tissue-scale complications [49]. STRAINS can be used alongside fluorescence labeling and deep tissue imaging techniques [50] to spatiotemporally assess diffusion and drug delivery in tumors. Collectively, these examples speak to the potential for implementing STRAINS to comprehensively study signal transduction *in situ* on the tissue scale for a wide range of systems.

Furthermore, STRAINS aligns with new techniques in multiplexed imaging and large-scale omics data collection in the push for spatially-resolved cell data. Recently developed methods such as PASTE can produce full tissue-scale renderings of transcriptomic data, enabling identification of gene expression and cell type within tissues [51]. Similarly, techniques like IBEX [52] or Cell DIVE [53] make use of immunofluorescent imaging to detect protein-level spatial organization of cells and tissues [54]. Further, spatially-resolved isotope tracking has recently been used to quantify metabolic activity in various tissues [55]. While most of these techniques capture data at a single time point, STRAINS enables real-time *in situ*, nondestructive spatiotemporal mapping and analysis of cell behavior in response to dynamic stimuli. Integrating STRAINS with such techniques would allow us to probe how any stimulus affects coordinated cellular responses on the milliseconds-to-hours timescale, resulting in patterns of, for example, gene expression, protein synthesis, energy utilization, or ultimately cell and tissue fate. As a new tool capable of simultaneously tracking multiple responses of thousands of individual cells and analyzing patterns of cellular behaviors, STRAINS provides insights into how events are coordinated in complex biological systems. By combining time histories of cellular responses with spatial maps of behavioral distributions, we have demonstrated that STRAINS can effectively make use of large datasets to study signal transduction and cell fate in the context of tissue injury and disease.

## Methods

The method consists of three main components to comprehensively study chondrocyte responses to strain: *in situ* fast confocal imaging, cell tracking and intensity extraction, and cell signal analysis.

### Impact-induced trauma to articular cartilage explants *in situ*

**Dissection.** Samples were sterilely dissected from the femoral condyles of neonatal bovids obtained from a local abbatoir (Gold Medal Packing, Rome, NY) within 24 hours of sacrifice. Cylindrical explants (6mm diameter x 3mm depth) were extracted with a biopsy punch (Fig 1a) and cultured for 24 hours at 37˚C, 21% $O_2$, and 5% $CO_2$ in low glucose media containing phenol-free Dulbecco's modified Eagle's medium containing 1% fetal bovine serum, 4-(2-hydroxyethyl)-1-piperazineethanesulfonic acid (HEPES) 0.025 mL/mL, penicillin 100U/mL, streptomycin 100U/mL, and 2.5mM glucose (Fig 1b). Special care was taken to ensure that the surface of the tissue was cut perpendicular to the depth of the cylinder to maintain uniformity of the strain field during mechanical testing.

Bovine synovial fluid (abbatoir derived, Lampire Biologics, Pipersville, PA) was applied to the joint surface to ensure smooth cutting and to lower the shear forces applied onto the tissue by the biopsy punch, to preserve as many chondrocytes as possible. Similarly, synovial fluid was applied to the blade of extra sharp razors used to bisect the sample. Samples were trimmed to 3mm thickness and bisected in a custom built stainless steel cutting jig, where the cartilage was submerged in a warmed PBS bath during the entire cutting process.

**Staining.** In order to measure cellular signaling and mitochondrial activity during impact-induced trauma, the tissue is fluorescently labeled with 3-color assay: (a) Calbryte 520 AM, a intracellular calcium flux assay to observe cellular calcium signaling (4μM, 1 hr incubation at 37˚C), (b) tetramethylrhodamine, a mitochondrial membrane potential indicator to observe mitochondrial polarity (TMRM, 10nM, 1hr incubation at 37˚C), and (c) Sytox Blue, a nucleic acid stain used to identify dead cell nuclei (1μM, 1hr incubation at 37˚C) (Fig 1b). The stains selected for the assay can be modified to reflect parameters of interest in the study.

Cylinders were bisected and mounted side by side on the back plate of a previously described confocal-mounted impactor, with the deep zone of the tissue adhered to the backplate of the impactor with superglue. Samples were submerged in a bath of Dulbecco's Phosphate Buffered Saline (DPBS) with Sytox Blue to ensure that cells dying over the course of the experiment were labeled (Fig 1c). Previous experiments have demonstrated the efficacy of the impactor system in inducing mitochondrial dysfunction and cell death.

**Injury.** The impacting device, which has been described and validated in previous experiments [21, 23], delivers an energy-controlled impact using a spring-loaded piston with an impacting tip consisting of an 0.8mm diameter stainless steel rod. The half cylinders are positioned so that one is centered on the impactor tip and the other is left as an unimpacted control. The impact lasts 5–10 ms, and produces a peak stress of approximately 1 MPa (Fig 1c). This replicates a superphysiologic loading rate which is characteristic of cartilage injury. However, it is not designed directly to replicate a specific loading pattern, but instead to expose the tissue within field of view of the microscope objective to a wide range of strains, allowing us to directly make connections between the mechanics of impact and injury to cellular responses.

**Confocal imaging.** The impactor was mounted to the stage of an inverted, spinning disk confocal microscope (3i Marianas) with a 10x objective, which allowed the capture of a 660μx 660μ(512 x 512 pixel) area. The site of impact, along with three surrounding locations (two adjacent sites on the articular surface and one adjacent site into the depth of the tissue), and two locations on the control sample were captured. Together, these imaging sites combine to

capture up to 1.2mm into the depth of the tissue, and 1.8mm laterally surrounding the impact site and up to a similar depth in the unimpacted control. Z-focus was centered on a depth 30μm away from the cut surface of the tissue to avoid imaging chondrocytes damaged during the sample preparation process. Additional sites can also be used, with limitations set by the scanning speed of the confocal microscope (Fig 1d).

In order to observe the peracute timecourse of events surrounding impact, we image continuously during and immediately following impact at a rate high enough to capture the cellular dynamics of injury. Cell calcium concentration is imaged in the green channel with 25ms exposure, a fast enough imaging rate to observe sub-second changes in chondrocyte calcium (Fig 2a).

A three-color staining assay is used to monitor the subsequent effects of impact in the longer minutes to hours timescale. Images were collected every 20 seconds sequentially at each site, with Calbryte 520 AM (green; 488nm excitation/499–553nm detection), TMRM (red; 561nm/563–735nm), and Sytox Blue (blue; 405nm/414–479nm). A slower rate of imaging is used to allow for longer exposure with the weaker red and blue signals, while simultaneously minimizing the effect of photobleaching (Fig 2b).

## Modified particle tracking

While the original Crocker and Grier algorithm [39] assumed a relatively constant intensity for the tracked particles, our cell fluorescence signal could vary significantly. Thus, to maintain a coherent track when fluorescence levels fluctuated, we interpolated between successfully tracked frames. We found that linear interpolation was reasonable when the distance between tracked frames was small, typically less than a cell diameter. Finally, once the tissue was relaxed and cells were nearly static in their position, we continued to measure fluorescence over long periods of time even when cells were not visible. This procedure captured tracks for ∼96% of the cells uniformly distributed over the entire tissue for time scales ranging from milliseconds to hours post impact.

The coordinates of each cell centroid in each frame were used to extract intensity data for each channel. In order to account for fluctuations in imaging and inhomogeneities within a cell, we averaged over a 3x3 pixel region surrounding the centroid pixel. The size of this region was chosen to ensure that data collection in one cell did not overlap with data collection in neighboring cells, or extend into the extracellular matrix. We then extracted the fluorescence intensity of each channel for each tracked cell over the entire experiment.

## Fluorescent intensity validation

Multiple controls were included for each of the fluorescent stains, in order to provide relevant comparisons with experimental data. Calcium staining controls were conducted with EGTA, a calcium chelator, and Thapsigargin, an endoplasmic reticulum calcium chelator, in 0mM $Ca_2^+$ media as a minimum, and with 10mM $Ca_2^+$ media as a maximum. Mitochondrial polarity controls were conducted with FCCP in the media, which fully depolarizes mitochondria, as a minimum, and incubation with oligomycin, which hyperpolarizes mitochondria, as a maximum. Nuclear membrane permeability/cell viability controls were conducted with ethanol in the media as a minimum, and incubation with P188, a membrane stabilizer, as a maximum.

## Background subtraction for confocal images

After tracking cells within the tissue, background subtraction is implemented. Impact-induced cartilage trauma can cause formerly cell-localized stains to leak into the extracellular matrix, locally increasing the background intensity of certain areas of tissue for some time. This causes

the background to be both non-uniform throughout the imaging frame and also changing with time. We address this problem by dividing each image into an 8-by-8 grid and subtracting the mean value of the twenty lowest pixel values within each region. For a 512-by-512 pixel image, this corresponds to the twenty lowest pixel values out of 4096, ensuring that no pixels within cells would be accounted for in this background subtraction.

When necessary, we used a moving average to smooth the data to remove high-frequency noise arising from fluctuations in the confocal images. The size of the moving average window is chosen to adequately remove high frequency noise without disturbing the shape or size of features of interest in the time series. For the time series data in our system, a window size of 20 frames is optimal.

## Details of feature extraction

To find peaks in the timeseries, we used the MATLAB FINDPEAKS function with slight modification. The 'minpeakprominence' parameter was set to 3, and a width-to-prominence ratio of 10 was used to filter out extremely wide peaks not considered calcium transients. An additional shape parameter was established by selecting for peaks with a set width to prominence ratio in order to filter out extremely broad peaks that should not be considered transients. Once the peak detection parameters were established for one data set, they could be used to identify peaks in subsequent data sets.

We identified sudden changes in the fluorescence timeseries data using the MATLAB FINDCHANGEPTS function. Here, we adjusted the residuals such that the fitting parameters identified either one or two changepoints per "step" in intensity, and additionally filtered for changepoints where the slopes before and after the identified point were significantly different, to distinguish from baseline drift. Similarly to setting the peak detection parameters, once changepoint detection parameters were established, they too could be used to identify steps in the fluorescence data of subsequent data sets. Notably, these features are highly customizable with user-adjusted inputs, and can easily be adapted to other systems.

## Full decision tree methodology

A visualization of the decision tree algorithm for our system is available in the S1 File. In this algorithm, individual cells are run through the tree one-by-one. We start with one of the most conditional arguments. In the following description, all numbers have arbitrary intensity units. Cells where the nuclear membrane permeability starts high (Fig 4a) are categorized based on their maximum blue value (greater than 7) and frame at which blue reaches maximum (less than 7). Then, the tree splits between cells that have blue changepoints and cells that do not have blue changepoints.

For cells that have blue changepoints, we ask if there are more than three changepoints to determine cells where the nuclear membrane permeability has multiple levels (Fig 4l). For cells with fewer blue changepoints, we determine if the mean difference between red and blue timeseries is greater than 20 (red being higher). If true, then the cell has polarized mitochondria but the nuclear membrane permeability still increases (Fig 4e). If false, then we determine if the cell has green peaks. If the green peaks are before the blue changepoint, then the calcium transient occurs and the nuclear membrane permeability increases (Fig 4h) and if they are after the blue changepoint, then the cell has calcium transients after the nuclear membrane permeablity is elevated (Fig 4g). For cells that have no green peaks, then we determine whether or not the blue changepoint occurs immediately after impact. Cells where the maximum value occurs after frame 120 and the blue changepoint after frame 100 are considered to have a late increase in nuclear membrane permeability, not immediately from impact. If true and there is

a green changepoint, then the calcium concentration drops and the nuclear membrane permeability increases (Fig 4f). If true and there is not green changepoint, then the nuclear membrane permeability increases late with no obvious trigger (Fig 4d). If the cell increases in nuclear membrane permeability immediately after impact, then we determine if it has blue peaks, which indicate an increase and decrease in nuclear membrane permeability (Fig 4c). If there are no peaks, and the range after the blue maximum is lower than 8, then the cell's nuclear membrane permeability increases and plateaus (Fig 4b). All remaining cells in this branch of the tree (has blue changepoints) are categorized as nuclear membrane permeability starts elevated (Fig 4a).

For cells that do not have blue changepoints, we first determine if the cell has green peaks. If yes, then the cell has physiologic calcium transients (Fig 4i). If no, we find cells where the range of blue values is greater than 10, indicating that the nuclear membrane permeability rises (Fig 4b). Then, if the range of blue values is lower than 10, we determine whether the mean of red values after frame 50 is lower than 6. This frame value is chosen to represent a significant enough time after impact where cells have stabilized to their state. For cells with a high red mean, they are normal cells with polarized mitochondria (Fig 4j), and cells with a low red mean have all low signals and depolarized mitochondria (Fig 4k).

Finally, in developing this decision tree we were able to make more informed decisions about how we manually classified cells. For example, the normal cells (Fig 4j), and the all low signal cells (Fig 4k) exist on a continuum. Both categories feature no calcium signaling and no elevated nuclear membrane permeability, with the sole distinction between the two being the intensity of mitochondrial polarity. While hand-sorted cells were judged to the best of our human efforts, there was no easily defined cutoff between the two categories. In programming the decision tree, we specifically defined a numerical cutoff for mean values of fluorescence which provided a more quantitative method of separating the two categories. By iteratively designing the decision tree and updating the manual classifications, we were able to produce more accurate classifications.

## Time series classifier parameters

Each classifer in the sktime library has its own model and training parameters. We optimized the parameters for our dataset by testing a range of values for each parameter. We found that while increasing the number of estimators or kernels may very slightly increase accuracy, the run time of training and testing these models increased significantly. We also tested other available parameters for these models, but have only listed relevant ones which noticeably affected accuracy.

For the Canonical Interval Forest (CIF) classifier, we used $n_{estimators} = 200$ and $n_{intervals} = 100$. For the Diverse Representation Canonical Interval Forest (DrCIF) classifier, we used $n_{estimators} = 200$ and $n_{intervals} = 40$. For the RandOm Convolutional KErnal Transform (ROCKET) Classifier, we used $n_{kernels} = 50000$. Finally, for the Arsenal Ensemble Classifier, we used $n_{kernels} = 10000$ and $n_{estimators} = 50$.

## Supporting information

**S1 File. Supplementary information.** SI contains detailed information on observations of the strain-dependent cellular response, a detailed figure on the decision tree structure, details of the tracking and intensity code, details of the GUI, details of the time series classification code, and application of the method to tracking and classifying worm neuron data.
(PDF)

**S1 Video. GUI Video.** Video demonstrating how the GUI works.
(MP4)

## Author Contributions

**Conceptualization:** Jingyang Zheng, Thomas Wyse Jackson, Lisa A. Fortier, Lawrence J. Bonassar, Michelle L. Delco, Itai Cohen.

**Data curation:** Jingyang Zheng.

**Formal analysis:** Jingyang Zheng.

**Funding acquisition:** Lisa A. Fortier, Lawrence J. Bonassar, Michelle L. Delco, Itai Cohen.

**Investigation:** Jingyang Zheng, Thomas Wyse Jackson, Itai Cohen.

**Methodology:** Jingyang Zheng, Thomas Wyse Jackson, Lisa A. Fortier, Lawrence J. Bonassar, Michelle L. Delco, Itai Cohen.

**Project administration:** Lisa A. Fortier, Lawrence J. Bonassar, Michelle L. Delco, Itai Cohen.

**Resources:** Jingyang Zheng, Lawrence J. Bonassar, Michelle L. Delco, Itai Cohen.

**Software:** Jingyang Zheng.

**Supervision:** Lisa A. Fortier, Lawrence J. Bonassar, Michelle L. Delco, Itai Cohen.

**Validation:** Jingyang Zheng.

**Visualization:** Jingyang Zheng, Itai Cohen.

**Writing – original draft:** Jingyang Zheng, Lawrence J. Bonassar, Michelle L. Delco, Itai Cohen.

**Writing – review & editing:** Jingyang Zheng, Thomas Wyse Jackson, Lisa A. Fortier, Lawrence J. Bonassar, Michelle L. Delco, Itai Cohen.

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
