## [Decision Letter · Decision Letter 0]

16 Sep 2022

PONE-D-22-21381STRAINS: A big data method for classifying cellular response to stimuli at the tissue scalePLOS ONE

Dear Dr. Zheng,

Thank you for submitting your manuscript to PLOS ONE. After careful consideration, we feel that it has merit but does not fully meet PLOS ONE’s publication criteria as it currently stands. Therefore, we invite you to submit a revised version of the manuscript that addresses the points raised during the review process.

We look forward to receiving your revised manuscript.

Kind regards,

Florian Rehfeldt

Academic Editor

PLOS ONE

Journal Requirements:

"The work was supported by the NIH National Institute of Arthritis and Musculoskeletal and Skin Diseases, Contract: 5R01AR071394-04, K08AR068470, R03AR075929, and The Harry M. Zweig Fund for Equine Research. Additionally, this work was supported by the National Science Foundation grants DMR-1807602, 5DMR-1808026, CMMI 1927197, and BMMB-1536463. Lastly, this work made use of the Cornell Center for Materials Research Shared Facilities which are supported through the NSF MRSEC program (DMR-1719875)."

"Funding: The work was supported by the NIH National Institute of Arthritis and Musculoskeletal and Skin Diseases (niams.nih.gov), Contract: 5R01AR071394 to LAF, K08AR068470 to MLD, R03AR075929 to MLD, and The Harry M. Zweig Fund for Equine Research to LAF and MLD. Additionally, this work was supported by the National Science Foundation (nsf.gov) grants DMR-1807602 to IC and LJB, CMMI-1927197 to LJB and IC, and BMMB-1536463 IC and LJB. Lastly, this work made use of the Cornell Center for Materials Research Shared Facilities which are supported through the NSF MRSEC program (DMR-1719875). The funders had no role in study design, data collection and analysis, decision to publish, or preparation of the manuscript."

Reviewers' comments:

Reviewer's Responses to Questions

**Comments to the Author**

1. Is the manuscript technically sound, and do the data support the conclusions?

Reviewer #1: Yes

Reviewer #2: No

2. Has the statistical analysis been performed appropriately and rigorously? 

Reviewer #1: Yes

Reviewer #2: No

3. Have the authors made all data underlying the findings in their manuscript fully available?

Reviewer #1: Yes

Reviewer #2: Yes

4. Is the manuscript presented in an intelligible fashion and written in standard English?

Reviewer #1: Yes

Reviewer #2: Yes

5. Review Comments to the Author

Reviewer #1: The paper illustrate a MatLab based code to follow multiple fluorescent channels over time for tissue scale samples. The code provided is fully documented and incorporates a GUI for non-programmers. The supplementary data is complete. The approach seems reasonable and the execution sound.

For minor revisions:

1. I want to address that the paper claims that the software can be used for a wide variaty of data. However, the autors show only one measuerement one bovine certilage of 5 ROIs and the GUI is customized to this measurement type.

Given the claim to be multi-purpose, exspecially lines 301 to 312, I would have appreciated to see the application to 1-2 other, probably openly available, datasets in the supplementary material.

2. The peak stress of "approximately 1 MPa" is mentioned and a "wide range of strains" accross the sample. As the stress and strain exerted seems crucial for the biological evaluation of such data, a proposal of strain/stress measurements or a proposal for theoretical strain/stress modelling for this data would improve on the method.

3. During the paper, the description of the experimental setup and locations in the tissue seemed unprecise and confusing. Examples are "outward from the injury site", "to the side/below the impact side"... "cells below the injury site". Mentioning relation to the surface level of the sample or the depth of the tissue would be helpful.

Reviewer #2: The authors present a software for visualization, analysis, and classification of (fluorescent) image time series. In particular, using confocal microscopy, the authors measure cellular Ca2+responses, mitochondrial potential, and nuclear membrane permeability of bovine cartilage samples in response to a brief mechanical impact. The impact site was imaged for 1.5 min with a frame rate of 40/s, and subsequently a region adjacent to the impact site and a region “behind” the impact site was imaged every 10 s for 4 h. A total of approximately 5000 cells were segmented from these images, and their fluorescent signals were tracked over time.

The presented software was developed to visualize and classify the data. The development of the software was motivated by the belief of the authors that “the sheer scale” of collected data was not amenable to typical statistical analyses. Where this belief comes from, what is meant by “typical statistical analysis”, or the scale of data beyond which the authors think typical statistical analysis might fail, remains unexplained.

The way the authors chose to analyze the data is to broadly categorize the response of each cell into descriptive signatures such as “Nuclear membrane permeability has multiple levels”, “Calcium transient, then no changes to nuclear membrane permeability”, or “Calcium concentration drops, then nuclear membrane permeability increases”. This classification was either based on elementary time series statistics, or on different multivariate time series classifiers. The spatial distribution of cells belonging to each of these categories was then computed. These maps show for example that the impact site has the highest number of cells with transient increase in membrane permeability. The row of cells behind the impact site show a calcium transient with subsequent increase in nuclear permeability, and the cells in the south-east corner diagonal behind the impact site show a calcium transient but no change in nuclear permeability. Some of these observations appear trivial, others are puzzling, but none of these observations contribute to any new insight, or to an understanding of how cells collectively coordinate their response to a mechanical stimulus.

Perhaps most unfortunate is that the empirical classification is not or only marginally meaningful. For example, it is unclear which categories are associated with cell death, as this was not specifically measured. Also, the speculation which categories represent fingerprints of Piezo activation versus TRPV4 activation is far-fetched and not helpful, as this was also not measured. Without an unambiguous meaning and interpretation on the basis of additional solid data, a classification of cell responses using descriptive signatures as provided by the STRAINS software has little scientific value.

6. PLOS authors have the option to publish the peer review history of their article (what does this mean?). If published, this will include your full peer review and any attached files.

Reviewer #1: No

Reviewer #2: No

---

## [Author Response · Author response to Decision Letter 0]

31 Oct 2022

Please see response to reviewers letter

---

## [Decision Letter · Decision Letter 1]

21 Nov 2022

STRAINS: A big data method for classifying cellular response to stimuli at the tissue scale

PONE-D-22-21381R1

Dear Dr. Zheng,

We’re pleased to inform you that your manuscript has been judged scientifically suitable for publication and will be formally accepted for publication once it meets all outstanding technical requirements.

Kind regards,

Florian Rehfeldt

Academic Editor

PLOS ONE

Additional Editor Comments (optional):

Reviewers' comments:

Reviewer's Responses to Questions

**Comments to the Author**

1. If the authors have adequately addressed your comments raised in a previous round of review and you feel that this manuscript is now acceptable for publication, you may indicate that here to bypass the “Comments to the Author” section, enter your conflict of interest statement in the “Confidential to Editor” section, and submit your "Accept" recommendation.

Reviewer #1: All comments have been addressed

2. Is the manuscript technically sound, and do the data support the conclusions?

Reviewer #1: Yes

3. Has the statistical analysis been performed appropriately and rigorously? 

Reviewer #1: Yes

4. Have the authors made all data underlying the findings in their manuscript fully available?

Reviewer #1: Yes

5. Is the manuscript presented in an intelligible fashion and written in standard English?

Reviewer #1: Yes

6. Review Comments to the Author

Reviewer #1: The authors have satisfactorily answered all my comments. I find STRAINS to be a useful tool for categorizing cells in tissue context. The method used is easy to replicate and the software is functional and well documented.

7. PLOS authors have the option to publish the peer review history of their article (what does this mean?). If published, this will include your full peer review and any attached files.

Reviewer #1: No

---

## [Editor Report · Acceptance letter]

29 Nov 2022

PONE-D-22-21381R1 

STRAINS: A big data method for classifying cellular response
to stimuli at the tissue scale

Dear Dr. Zheng:

I'm pleased to inform you that your manuscript has been deemed suitable for publication in PLOS ONE. Congratulations! Your manuscript is now with our production department. 

Kind regards, 

on behalf of

Dr. Florian Rehfeldt 

Academic Editor

PLOS ONE